# Church Building as a Secular Endeavour: Three Cases from Eastern Germany

Agnieszka Halemba

Institute of Archaeology and Ethnology, Polish Academy of Sciences, 105, 00-140 Warszawa, Poland; ahalemba@iaepan.edu.pl

**Abstract:** Political secularism can be defined as a kind of political philosophy that sees the secular state as setting the terms of encounter between the secular and the religious. However, religion and religious organisations are not necessarily seen as oppositional to the secular state; there can be myriad forms of coexistence between secular and religious authorities. The argument forwarded in this article is based on ethnographic research focussing on the presence and social significance of religious materiality in the region considered to be one of the most secularised worldwide—the north-eastern part of contemporary Germany. I investigate the strategies of actors socially recognised as either religious or secular towards each other, looking at how secular actors assign a place to religious symbols, materiality, theological concepts, organisations, and communities; on the other hand I investigate strategies that religious actors adopt in a context of political secularism. Even if political secularism presupposes supremacy of the secular state over religious actors and the right of the former to make legally binding decisions concerning the latter, those religious actors are not passive—they react to secular initiatives and they try to carve for themselves a space in a public sphere, while at the same time the secular or rather nonreligious actors mobilize religious elements for a variety of reasons.

**Keywords:** political secularism; eastern Germany; church buildings; reconciliation; secular/religious divide

## 1. Introduction

Secularism as a political project and as an ideology can appear in many guises, but most often, it encompasses a vision of religion as a problem. Elisabeth Shakman Hurd (2009) started the introduction to her book on the politics of secularism and international relations precisely with this statement, in capital letters: religion is a problem. In this formulation, secularism is about the struggle to either keep mobilising forces of religion at bay and under control or to assign certain rights to religion so it does not become a problem (again). However, Verkaaik and Arab (2016) pointed out that although the theories concerning both secularisation as a process and secularism as an ideology, principles of governance and political project provide immensely important and useful tools for researchers, they are constructed abstractions that do not and cannot be expected to account for the diversity of actual practices. Therefore, those authors encourage the use of ethnographic case studies to provide new insights into existing secularisms. Is religion a problem for secularism as a political project in practice? Is it maybe a partner in the discussion or, rather, a potentially useful addition to the political toolbox?

This paper takes up the invitation to look at and reflect on political secularism by means of ethnographic case studies conducted in Brandenburg, the region of Germany where both secularism as a statecraft principle (Casanova 2009) and secularisation as a process, especially in terms of a decline in belief and belonging to religious organisations (Casanova 1994), seem to be obvious and advanced. The case studies presented here show that the differentiation and separation of secular from religious in the public sphere is a complex issue, inviting ethnography-based analysis.

My argument takes as a starting point an assertion expressed most prominently in Talal Asad's works (Asad 1993, 2003), that the history of relations between the secular and the religious is also the history of their mutual definition. It is not possible to form a universal definition of religion, just as it is not possible to form such a definition of the secular (Asad 1993, 2003). Therefore, I look at ethnography as a way of presenting particular situations that take place under the conditions of what I call "confident political secularism". After Berlinerblau (2017, p. 94), I take political secularism to be a kind of political philosophy and practice in which the secular state grants itself the right to unilaterally define the rules of the relationship between secular and religious; there is, therefore, still a changing and historically specific definition of both, as in Talal Asad's formulation, but the defining process is not mutual.

I use the term "confident political secularism" to describe the ideological background of those situations where the main principle of subjugating the religious to the secular in the defining process goes virtually unquestioned by all public actors—religious, secular, or nonreligious. In this particular article, I focus on situations in which the main actors consider themselves and are considered by all participants of the given social conversations as native, majoritarian, or historically dominant. Those who are considered as minorities or newcomers, or are socially classed as alien, are on the fringes of the discussed negotiations. This makes the described cases different from most of the available ethnographic studies of secularism in Europe, which mainly concern religious groups, organisations, or citizens who, being in a minoritarian position, negotiate their needs in the public sphere (e.g., Modood 2019). The presence of Islam, Judaism, and other religious denominations is also an important factor in the cases that I ethnographically describe, though they are also on the fringes of discussions. In the present analysis, I focus on the practices and discourses of the dominant actors.

In confident political secularism, religion occupies a clearly subjugated but fairly neutral position; it is a subject of decisions and discussions taking place in the public sphere, where religious actors can also participate, but where the dominance of secular governance is accepted and goes virtually unquestioned. Religion and religious organisations are not seen as being in competition or in opposition to the secular state; instead, they can be seen as useful for secular actors, state and nonstate alike, as they provide a set of tools that can advance various agendas, not necessarily directly related to religion or even in line with goals of religious organisations and groups.

Most importantly for this article, at any historical moment when the secular and the religious are differentiated in a particular way, we see the production of material objects socially recognised as religious. In the case of Christian religions in Europe, these are, most importantly, church buildings and chapels, but also such elements of the public space as bells (including their sound), crosses, paintings, and sculptures presenting themes with religious connotation. These material objects stay in the public space and often also retain their significance when relations between the religious and the secular change. Their often powerful and even dominant presence is recognised as a value and resource by a multitude of actors.

The ethnographic analysis presented in this this article is aimed at showing how elements socially recognised as religious are mobilised for secular goals and how religious actors find a place for themselves in a confidently secular public sphere. Political secularism presupposes that the state can also define the scope of its own jurisdiction with regard to religious organisations in a way that transfers certain spheres of life and certain tasks outside its own control and in the hands of religious organisations. Religion can be a welcome contributor to the public sphere, but on terms decided by the secular authorities. Such an understanding of relations between religious and secular can be found where religious actors are invited to participate in public discussions, but the terms of the discussions are delineated by secular authorities, as described in the later works of Juergen Habermas (2006). Although the concept of political secularism is mainly used to refer to the relationship between secular state power and religious power, it can be assumed that the

results of this political philosophy are visible in the public sphere in general with regard to the relationships between all kinds of social actors considered and recognised as secular, religious, or nonreligious.

A notion of the nonreligious is important for my analysis and is defined here in a particular way, proposed by Johannes Quack (2014). According to him, as nonreligious, we can describe those social actors who see themselves as situated outside the religious field but are at the same time very much interested in the religious one. Nonreligious actors are part of the secular, understood as differentiated from the religious (Casanova 2009) in terms of self-definition. However, they mobilise religion either by assigning it a positive symbolic, practical, or social role, or by making it into the enemy, as an opponent and challenge to the secular. Religion for them is a tool to achieve goals in the secular public sphere. Such actors are the main protagonists in my analysis.

Another notion that is crucial for my argument is one of materiality. The so-called material turn has been one of the most important and fruitful avenues in the anthropology of religion in recent decades (Houtman and Meyer 2012). The main topic of those works is the role of bodies, infrastructures, and other forms of materiality in making religious experience possible and in maintaining forms of religious practice, learning, and belonging (e.g., Meyer 2009; Engelke and Tomlinson 2006). In my analysis I take a different angle: I start from an observation that materiality socially recognised as religious is important for many actors who situate themselves on the secular side of the religious/secular divide, but through their interest in religious materiality become nonreligious in Quack's understanding. Church buildings in contemporary Europe are socially recognized as religious places even though the relations between religious and secular are now different than at the time they were built. Such buildings can be appreciated and identified as important for a variety of reasons that transcend religious convictions and devotional practices. However, it seems that although the religious function of such places might have faded into the background, the fact that those places are socially recognised as religious has consequences for public debate (see also Halemba 2022).

## 2. Fieldwork

The particular argument advanced in this article is based on ethnographic research that started in 2020.[1] It is focused on the presence and social significance of religious materiality in the region considered to be one of the most secularised in Europe, the north-eastern part of contemporary Germany, formerly a part of the German Democratic Republic. The project is composed of three case studies. Firstly, I work with people involved in renovating and sustaining Protestant village churches in the central and northern parts of Brandenburg country. This includes interviews with members of NGOs, people living in villages where the churches are located, and local pastors, as well as representatives of the central and regional church administration, especially those working for the building department.

The second case study concerns the initiative of rebuilding the Garrison Church in Potsdam, the capital city of Brandenburg. Here the focus is on an analysis of public debates as presented in the local press, leaflets, documents, books, and official public hearings and during events organised by various involved actors. This is supplemented by interviews with the main protagonists and (passive) participation in many public events and debates concerning this endeavour.

The third case study is only briefly introduced. The research work has been conducted by Piotr Winiarczyk, a student at the University of Warsaw who is working with me on this project. It concerns the activities of the Reconciliation parish, which has as its basis a chapel that is part of the Berlin Wall Memorial at Bernauerstraβe. This research in progress involves participating in activities organised by the Reconciliation parish and conducting interviews with parishioners, other people involved in these activities, and staff at the Berlin Wall Memorial.

All of those locations are in a region of Germany that is widely regarded as one of the most secularised in Europe. They are connected by a powerful presence in the public space

of materiality, which is socially recognised as religious. The church buildings influence and even shape the cityscapes, despite the fact that religious worship is fading, and some locations do not have a Christian religious community of any considerable size at all that uses the buildings on a regular basis. Still, the buildings themselves make religion quite present in public space; moreover, through this presence, religion is evoked, and religious actors participate in the public sphere in a more general sense: their presence is one of the factors ensuring the important place of religion in public discussions.

The aim of this research project is to investigate the strategies that actors socially recognised as either religious or secular use towards each other. I am interested, on the one hand, in looking at how secular actors assign a place to religious symbols, materiality, theological concepts, organisations, and communities; on the other hand, I look at strategies that religious actors adopt in the context of confident political secularism. The dominance of political secularism as a political philosophy does not always mean a reduction in the role of religion in the public sphere. On the contrary, it sometimes occurs that religious elements are invited, as it were, to participate in public life by actors who see themselves as not involved in the religious field.

### 3. Religion and the State in Former GDR

Eastern Germany is considered one of the most secularised regions of Europe in terms of religious membership, but also declaration of belief and individual religious practice (Pollack 2002; Pollack and Pickel 2000). In recent years, the number of believers has diminished, especially in the Catholic and Protestant Churches,[2] the two organisations which were still called *Volkskirchen* in the first half of the 20th century. This term literally means "people's churches" and denotes ecclesiastical organisations with particularly strong links to and a significant role in and for society, which goes beyond religious rituals and other devotional practices. During my research in Brandenburg, I often heard from pastors and believers that the Evangelical Church in Germany (*Evangelische Kirche in Deutschland*, EKD) lost its role as a *Volkskirche*, as at present it does not have broad popular resonance and recognition. Instead, the ecclesiastical organisations are recognised not only legally but also by those who live in small towns and villages as *Landeskirchen*, literally "state churches"; that is, formal organisations that have a defined administrative area, largely (though not always) coinciding with the borders of the federal states and linked to the state through multiple administrative channels. Hence, while *Volkskirche* refers to Church that is part of the daily lives of the majority of inhabitants of a region and an important factor in their self-identification, *Landeskirche* is primarily used to refer to administrative unit and indicates the existence of organisation that has as its main partners not the people but the state and its apparatus.

This transition from *Volkskirche* to *Landeskirche* provides a good description of the situation in eastern Germany. In Potsdam, the capital of Brandenburg, in 2019 82% of inhabitants did not belong to any religious community.[3] In the state of Brandenburg at the end of 2020, only 17.5% of the population belonged to either the Catholic or the Evangelical Church,[4] with many people leaving both churches every year. Usually, this situation is understood as a legacy of the German Democratic Republic's atheistic position and successful propaganda. However, some authors point out that secularising tendencies were already evident in this region of Germany earlier and are linked to 19th-century industrialisation and socialist movements (Froese and Pfaff 2005). In eastern Germany today, there are many people who, according to Monika Wohlrab-Sahr, "are often stubbornly secular",[5] claiming that religion is simply a non-issue for them. "People have forgotten that they have forgotten God"[6] is a sentence attributed to the former head of the Evangelical Church of the Church Province of Saxony, Bishop Axel Noack. It has been repeated many times in speeches, articles, and popular books as an adequate description of the relative insignificance of religion in the private lives of many people in eastern Germany.[7]

It does not mean, however, that religion does not play any part in social life. There might be a crisis of the EKD as *Volkskirche*, but the EKD as *Landeskirche* is still very strong.

The two main Churches in Germany, Evangelical and Catholic, together with Jewish religious communities, receive contributions from their members through the state tax system and are also partly supported by state money in various forms, most importantly the so-called *Staatsleistungen* (state benefits), which originated in agreements between the German states and the Churches dating back as far as the beginning of the 19th century, adding up to hundreds of millions of euros every year.[8] There are also subventions paid by the state and linked to church-run services in the form of schools, kindergartens, counselling centres, retirement homes, and hospitals. The Churches are also important land and estate owners, with considerable assets and leasing strategies that influence local politics. Despite the shrinking numbers of believers, the Evangelical and Catholic Churches are also powerful and influential organisations in eastern German states. Moreover, as Peperkamp and Rajtar (2010) observed, communities of believers may be numerically small, but they are very active, influencing social life in many ways.

There is also the visible part of religion, especially significant for the present argument. Church towers define the skylines and landscapes not only of east German cities but also the countryside. Brandenburg is the German country with the greatest density of Evangelical Church buildings in relation to the number of church members.[9] There is a growing number of research projects that address the issue of the post-religious life of church buildings (Halbfas 2019; Keller 2016; Siegl 2018).[10] Interestingly, many civic initiatives concerned with the renovation or maintenance of church buildings where services are rarely held due to lack of both believers and pastors are situated in eastern Germany, especially in Brandenburg and Mecklenburg–Vorpommern.

## 4. What Is a Church in a Small Village?

*Alte Kirchen Berlin Brandenburg* eV. is an association established in Berlin in the early 1990s by a group of people who, in one capacity or another, professionally or out of personal fascination, were interested in restoring church buildings. There were about 35 initial members from both East and West Germany. Today, the association has approximately 650 members[11] and, over the last 30 years, has been involved in dozens, if not hundreds, of projects.[12] The attitudes of the members to religion and their relationships with religious organisations vary. They are also not a decisive factor when it comes to membership. In this sense, it is a nonreligious association focused on materiality, which is socially recognised as religious, for which the Church, in this case, most prominently the *Evangelische Kirche Berlin–Brandenburg–schlesische Oberlausitz* (EKBO), is one of many partners in negotiations over the fate of church buildings in Brandenburg's villages.

The objective of the association is to support local initiatives aimed at preserving and/or restoring church buildings that survived the times of the GDR in rather dismal conditions. During the GDR era, church buildings were sometimes renovated (Dohmann et al. 1964); still, many of them were in a rather dire state at the end of the 1980s. At the beginning, the association focused on helping local initiators to get their restoration projects off the ground, or in some cases actively initiated projects, trying to mobilise local communities to act. More recently, the association has supported the development of the so-called "concepts of use" for renovated churches; while one aim is to retain the religious function of those buildings as much as possible, everyone seems to agree that religious use does not exhaust the potential of those places and, moreover, cannot be the only use in terms of economic sustainability because of the dwindling numbers of believers and pastors. Therefore, there is a need to think about possibilities for additional use. Most prominent are suggestions to use the buildings as concert or exhibition halls, but there are many other creative solutions.

In many villages, it is primarily not the parish councils but secular associations that carry out such projects. In a considerable number of cases, parish councils have been explicitly uninterested in restoring and maintaining certain churches. Nowadays, the borders of a parish extend beyond the borders of one village and can encompass several settlements. Therefore, within the territory of one parish with a few hundred members and

fewer than a hundred practicing believers, there can be several church buildings that are the responsibility of the relevant parish council. Moreover, in such regions as Uckermark in Brandenburg, many of those buildings have an official status as architectural monuments (*Denkmal*), having been built as early as the 12th century, which means that renovation has to be carried out under the supervision of state conservators, and the scope of changes and adaptations that can be made to the interior and exterior of buildings based on local needs is limited. Being responsible for several buildings that demand specialised attention is often too great an economic challenge for a parish with a small number of members who, furthermore, do not need those buildings for religious worship.

In this situation, many parishes, often with support of EKBO as an organisation, have considered privatising their church buildings or closing and abandoning some of them altogether and waiting for the forces of nature to do their work. The building department of the EKBO even decided to divide their buildings into four categories. Category A encompasses those that are considered indispensable to religious activities. In such cases, the local parish can count on the support of the Church administration to maintain them. Category B buildings are still in use and can also be supported. Category C and D buildings are of no interest to the Church, and in practice, the parish community cannot count on any support for their maintenance and restoration.

Many people in small towns and villages who are not church members are keen to get involved in efforts to preserve church buildings. Moreover, I have repeatedly encountered people who do not see themselves as religious asserting that churches should be preserved as recognisable religious buildings not only in their form but also in their function, by providing religious services at least a few times a year. In general, such people are also against privatisation of church buildings, a practice that is quite present in the western part of Germany and in other countries of western Europe. For the nonreligious inhabitants of eastern Germany, those buildings should remain in the public domain and any use beyond religious worship should be treated as an extension or addition only.

According to Protestant theology, unwanted buildings can be, under certain conditions, privatised, as there are no ritual consecrations of buildings in the Evangelical Church. Still, in practice, churches are not treated as ordinary buildings. Since 2005, the EKBO has published guidelines for the use and reuse of churches, in which it is stipulated that privatisation can be an option under certain conditions.[13] However, in practice, privatisation is very rarely seen as a solution by people who live next to or are interested in a particular church. One of the members of the association, himself a religious person, expressed this in the following way:

> We have always been against churches being privatised, and we want them to be used for public benefit. And the best thing, in my opinion, is if services continue to take place even if there is only a small congregation and religious services take place maybe only three, four times a year. But that's the thing, that the purpose is maintained and that there is an extension of it, like concerts or something similar. I know that churches cannot be used profitably in a market economy. No matter how much effort you put in. And this should not be seen as a disadvantage, but as an opportunity. These are the last remaining public spaces [in a village]. And if three services a year and two or three concerts take place there, that's enough. These buildings have a justification in themselves, through their history, through their religious history, through their cultural history, the history of the village. And they are, as I said, the last public spaces we have. They are the oldest buildings in the village. And that is also the reason why people keep coming together, also those ones who no longer want to have anything to do with the Church as an institution. But they say it is important that the church is repaired, that it is open. That is important to us. So there are over 300 associations in Brandenburg that look after churches. And I guess at least two-thirds of the members are not church-affiliated.

There are several important aspects in this quotation. My interlocutor was against the privatisation of churches, a sentiment expressed very often during my research. However, interestingly, he also insisted that churches should not be totally turned into cultural or concert halls and the religious function should be retained. Over the course of my research, it became clear to me that many people see religious services as a way of ensuring that the church stays accessible to all, believers or not. Religious practice is seen as insurance for public access.

One has to remember that in the years after the unification of Germany, many institutions with not only economic but also social functions disappeared from eastern German villages and smaller towns: shops, bars, restaurants, community houses, schools, and kindergartens were closed or moved to bigger settlements. The remaining buildings were abandoned or sold for symbolic money to private owners. For many people, church buildings are seen as the last public places in their villages, and religious services are seen as a weapon against privatisation. If a place is seen as having, among others, a religious function, there is the hope that it will not be privatised. It therefore happens quite often that in renovated buildings, those few remaining religious services a year also attract people who are not believers and not members of the church, especially if a particular church is seen as being directly threatened with privatisation.

Churches are also seen as places that can bind people together. The head of a local association, himself a declared nonbeliever, told me about the church building in his village:

> The church was quite overgrown, there were trees as tall as a man. It was like a Sleeping Beauty's castle, and someone wrote in the newspaper that the church had been awakened from its slumber, and the whole village was on its feet, working on the roof, atheists and Catholics as well. None of that mattered at all and this village itself came together again. Afterwards, we held a village festival, where the village was really complete and where we realised how long it has been since we sat together as a village community and celebrated, and how nice it is that we can talk to each other again.

Although the church as an organisation is not a *Volkskirche* anymore and the numbers of believers are falling, religion is still present in public spaces, as religious buildings still have an important social function. Church buildings can be mobilised for a variety of purposes, and it is their religious function that enables such mobilisation also for nonreligious actors. There is also a minority among people involved in the care of village churches who see them not only as protection against privatisation and the difficult or unjust effects of post-socialist transformation, but as defenders of the Christian identity of Europe, and in this way as a tool of exclusion. Still, in either case, religious materiality is mobilised by secular actors to address issues that go beyond the religious field.

## 5. Is a Garrison Church a Religious Place?

Despite the fact that there are many church buildings in villages and towns of Brandenburg that are in obvious need of repair and maintenance, and that hardly any of them are full of believers on a regular basis, in the city centre of Potsdam, there is a growing church tower. This tower, the building of which started in 2016 and should be finished in 2024, if not earlier, is a part of a project for a reconstruction of the Garrison Church, originally built between 1730 and 1735 by an order of Friedrich Wilhelm I, the "Soldiers' King" in Prussia,[14] and partly destroyed at the end of the II World War and blown up in 1968 on the order of GDR authorities (see also Halemba 2023).

This church has a complex history. For the present argument, it is enough to say that it was used for celebrating Prussian statehood and also played a role during the Nazi time in Germany (Grünzig 2017). Most importantly, some events of the so-called Day of Potsdam, 21 March 1933, took place partly in or in front of this church. On this day, the new German parliament, with Adolf Hitler as the new chancellor, was formally opened there; the *Reichstag* in Berlin had already burnt down, and Potsdam was selected as the place for

the formal ceremony. There is a famous photograph showing Hitler, dressed in civilian clothes, bowing his head while shaking hands with Paul von Hindenburg, dressed in full military uniform with medals and insignia. The new chancellor, the leader of the National Socialists, and the old president, representing the Prussian military elite, are standing in front of the Garrison Church in Potsdam. This photo is very important for discussions around the rebuilding of this place and has been interpreted and reinterpreted in a variety of ways. This is one of the most hotly debated issues in the city, and the construction has both staunch supporters and acrimonious opponents.

Importantly for the present argument, the idea to rebuild this church did not come from the EKBO as an organisation; the initiator of the reconstruction was Lieutenant Colonel Max Klaar, the head of the *Traditionsgemeinschaft Potsdamer Glockenspiel* (TPG; Traditional Association of the Potsdam Carillon), a society formed by a group of professional soldiers devoted to the protection of Prussian military traditions. Klaar managed to gather some initial, quite considerable funds as seed money, and approached the EKBO with his plan for rebuilding. He wanted the church to be a place of military tradition and national identification (Oswalt 2020), and he saw an important role for religious rituals and spiritual guidance in this process. Klaar expected not only that the church building would be returned to Potsdam but also that a religious work with military personnel, understood in very conservative terms, would be reinstated. The TPG expected from the Church a legally binding renunciation of practices such as church asylum and a ban on the blessing of homosexual couples and feminist theology, and to refrain from counselling conscientious objectors to military service (Chronologie des Wiederaufbaus 2021).

For the Church as an organisation, this was a tricky situation. For many people in Potsdam, the restoration of this church building was seen as part of a larger endeavour to restore the baroque (and Prussian) character of the city in general.[15] The Garrison Church was perceived as an important architectural landmark. Those people, including some important players in the local and national political arenas, saw this rebuilding as a necessary step towards the restoration of the city in general. There was a strong pro-rebuilding lobby in the city, who did not share Klaar's and the TPG's militaristic ideals, but nevertheless wanted the church to be rebuilt and used. In this situation, the EKBO decided to take part in the rebuilding process, despite repeated declarations that the building itself was not needed for religious worship. The EKBO rejected Max Klaar and his money and committed itself to supporting the rebuilding of the church as a place of reconciliation open to all.

This involvement by the EKBO was also welcomed by secular actors. The rebuilding of this church, or even only its tower, is an expensive endeavour; at the time of writing, the cost was already at EUR 44 million counting for the tower only. Because of the difficult history of this church, and because of the high cost, it is difficult to argue for rebuilding it only in terms of its architectural beauty. There was therefore a need to find a purpose for this edifice that would go beyond religious services, and at the same time be compatible with the general image of the EKBO as a religious organisation and acceptable as a legitimate use by the general public. It also seems that nonreligious supporters of the project viewed the EKBO's involvement as a way to ensure or publicly show that the difficult history of the Garrison Church would not cast a shadow on its future. It was the Church's task to make sure that this building would not be associated with the dark side of German history but would rather be seen as an important landmark and tourist magnet. This was especially important, as the opponents of rebuilding repeatedly stated, and state to this day, that this church has the potential to gather right-wing and revanchist groups, glorifying the Prussian past or even relativizing the period of national socialism. Religion has been mobilised by those who support rebuilding this church to weaken such accusations.

With respect to this task, the EKBO undertook at least two actions to make sure that a religious dimension will remain present at the site, that the Church as an organisation will retain at least some control over the significance of this church, and that the image of the Church presented at the site will be broadly accepted by secular society and provide

an adequate representation of religion and the Church. The first action was to prepare a conceptual framework for the use of the building, including some concepts that, while having roots in Christian theology, are also important for secular audiences; the second action was to organise a special nonterritorial parish, whose pastor and council would be responsible for the practical application of the concept of use. These two tasks guided much of the EKBO's involvement in the project between 2001 and 2020.

The most important notion that has been advanced as a guiding principle of the new conceptual framework for the rebuilt Garrison Church is one of reconciliation (for a detailed discussion, see Halemba 2023). This is a notion that is present in both theology and in the political arena. In Protestant theology, reconciliation refers to resolving the disruption between God and people through faith (Klein 1999). Whereas in Catholicism it is related to the sacrament of penance, in various Protestant denominations, reconciliation refers primarily to a personal relationship with God and to finding oneself through this relationship. Reconciliation is a fundamental aspect of Christian soteriology; it can be interpreted as a private matter for a given believer, a matter of "inwardness", something to be achieved in one's relationship with God.

Still, particularly since the Second World War, reconciliation has become a term that is commonly used in international diplomacy; it has emerged as a widely acclaimed goal of international politics, although there are, of course, many different and contradictory ways in which the notion is understood and implemented in practice (Malley-Morrison et al. 2013; Ash 1999). Reconciliation has been approached both as a goal and as a process and has generated an abundant body of literature in political science, focused mainly on analyses of particular reconciliation processes and the conditions necessary for achieving the peaceful coexistence the term implies (Malley-Morrison et al. 2013; Murphy 2010; Prager and Govier 2003; Quinn 2009). It is, therefore, a politically salient concept upon which religious authorities can claim to have special expertise, because of its theological roots, but at the same time a very salient concept for secular actors.

It is especially important in Germany, where the concept of reconciliation has been potent in the process of *Vergangenheitbewältigung*, defined by the *Duden*, a German dictionary, as "a nation's confrontation with a problematic period of its recent history, in Germany especially with National Socialism."[16] There is an enormous amount of literature on reconciliation, as both a concept and a process, including works that analyse particular cases (with German–French reconciliation often serving as a master example of the process; see Defrance 2019), as well as more general theoretical and political works. As Timothy Garton Ash has argued, in Germany, it is almost impossible to question the widely accepted claim that those who forget cannot reconcile and are condemned to repeat past crimes. Remembrance is seen as a moral obligation of both individuals and states (Ash 1999, p. 296).

This focus on reconciliation was presented in three subsequent "concepts of use" (*Nutzungkonzept*) prepared first by Church-organised committees and later by the Garrison Church Foundation (*Stiftung Garnisonkirche*), established by the EKBO in 2008 to take care of all matters related to the rebuilding. As a prime executor of this conceptual framework, a small nonterritorial parish of the Cross of Nails was established in 2011 in a provisional chapel behind the building site. For the next nine years, this community organised all kinds of events, both religious and nonreligious, many of which had reconciliation as a theme. It is important to note that the community is officially a member of the international Community of the Cross of Nails, established in Coventry after the Second World War as a network devoted to reconciliation. This link to the internationally recognised network contributes to the positive image of the place and the rebuilding initiative in general. Since the end of 2020, however, when the pastor of the Cross of Nails community retired, several disputes and events occurred that significantly changed the power relations within the supporting organisations. As of January 2023, the main organiser of events in the chapel behind the building site is the Supporting Society for the Rebuilding of the Garrison Church (*Fördergesellschaft für den Wiederaufbau der Garnisonkirche e.V.*), which is an association of

people who support the rebuilding for any "religious, spiritual and cultural-historical or urban planning reasons".[17]

The building site of the Garrison Church is a place where nonreligious actors, who are the most active ones, set the tone for what is happening. On many occasions, I witnessed members of the Cross of Nails parish community expressing their concern and determination that the needs of the group as a religious community are taken into consideration in the planning process. They understand that for many supporters of this endeavour, including the local representatives of the state, this church building is important for reasons that go far beyond religious use. It should become a tourist attraction as an architectonic gem, an aesthetically pleasing part of the urban space or a site of memory politics. The religious community is present there because of the very materiality of this building, as its shape is recognised as being characteristic of a church, and because religion makes possible certain avenues that enable justification and social acceptance for this project.

### 6. How to Be Important Yet Invisible

The use and shape of the Chapel of Reconciliation, located on the grounds of the Berlin Wall Memorial on Bernauerstraße, present a stark contrast to the cases introduced in previous sections. Although the Berlin Wall Memorial is dotted with religious symbolism, was erected partly on church grounds, and has historical and present links to active religious communities located nearby or onsite, those religious dimensions do not dominate the perceptions of visitors and tourists, who are interested primarily in the history of the Berlin Wall (see Harrison 2019). The Berlin Wall Memorial is perceived as a site of secular memorialization, with focus on a divided city and victims of Communist violence.[18] Still, the religious communities had and still have an important role in establishing and maintaining what appears to be a secular and state-run site of remembrance.

When the Berlin Wall was built in 1961, the neo-Gothic Church of Reconciliation was trapped in the middle of a guarded border strip. Subsequently, it was closed to the faithful for more than 20 years and its tower was used for observation by border guards. In 1985, the church was blown up and the remains were removed. After the fall of the Berlin Wall, the land upon which the church had stood was returned to the Church of Reconciliation, located at that time in the western part of the city. Its pastor, Manfred Fischer, was instrumental in establishing this site not only as a religious place, but more importantly as a memorial site (Harrison 2019). At the time, there was limited readiness to commemorate the Berlin Wall, especially among the population of East Berlin, as many people wanted the Wall to disappear without a trace from the city landscape as swiftly as possible. Still, owing to the efforts of Pastor Fischer and several other people, the Berlin Wall Memorial was established. In 1999, a Documentation Centre was opened in the Reconciliation parish house and a year later the Chapel of Reconciliation was consecrated.

The Chapel of Reconciliation is a very interesting architectural structure. Few people perceive its external shape as unambiguously ecclesiastical; above all, the chapel has no tower, and the cross is only visible on the façade in a way that makes it appear as a mere play of shadows. The original cross that was on the tower of the former church lies on the ground near the chapel. As a result of the fall during the detonation, it is bent and damaged; it was decided to leave it on the ground in this condition. The oval building is made of clay mixed with debris from a church that was blown up and is enclosed by an openwork wooden structure. It gives the impression of a structure that is lightweight and blends in rather than dominating the surroundings. The church bells are hung in a frame outside the chapel. The chapel is placed where the Reconciliation Church had been, but its dimensions are much smaller; the ground plan of the old church is marked around the chapel and serves as a church square. Inside the chapel, on the wall, hangs a woodcut with the scene of the Last Supper, which bears many traces of damage—Jesus' face is cut off, for example. It was brought out of the old church before the detonation. According to the current pastor, this woodcut was destroyed by soldiers using the church as an observation post. It was decided to leave the damage as it is, without repairing it.

The difference with the rebuilding process of the Garrison Church is glaring. In Potsdam, the tower will decisively change and mark the skyline of the city, but also, the way in which the religious community operates is very different. At Bernauerstraße in Berlin, the pastor and the parish community have very important roles: the pastor doubles as a guide at the site, and the parish members are responsible for commemorative events that take place regularly. Moreover, many parish activities are focused on places and issues that are not straightforwardly recognised as religious activities, and some active members of the parish and many of those who take part in parish-organised activities are declared nonbelievers. For example, behind the parish is a community garden. To receive a plot, one does not have to be a member of the church or the parish; it is open to everyone based on the availability of plots. Still, the pastor and parish members explicitly say that this is the place where, for them, the most important form of religious activity takes place: bringing people together. The pastor even uses the term "church" while talking about the garden. It seems that in their view, religious activity is not clearly separated from secular activity.

Even more significantly, this is also the position of the secular administration of the Berlin Wall Memorial, who explicitly accept religious actors and invite them not only to participate in but also to have a decisive voice on many activities at the site. Moreover, some employees of the memorial centre are members of the parish, despite the fact that they are declared nonbelievers. In interviews, they said that they wanted to be part of this particular parish because of the many kinds of inclusive activities that take place there, which makes explicit the one important message of the memorial in general: religious and secular life are intertwined and religious materiality melts into the dominant secular space. The Reconciliation parish is a member of the Cross of Nails international community as well, as is the small parish behind the Garrison Church building site in Potsdam. However, this membership is not prominently displayed. The Cross of Nails hangs at the entrance to the chapel, yet the role of the international association was neither raised in interviews nor is it mentioned on the parish website[19]. The parish was probably invited to join the Cross of Nails network because of its special location; however, its activities have a deeply rooted local character.

## 7. Conclusions

As cases of "confident political secularism", we can describe situations where the religious sphere is subjugated to the secular state but also fully accepted as an agent in the public sphere and public space. Religious materiality can be visible or hidden, and religious actors can be invited to run and manage the day-to-day activities in a given area or be instrumentalised for specific secular political needs, but religion is not often seen as a problem. Religion, in this case, is understood primarily as a complex of social practices, organisational solutions, behavioural patterns, and material objects that can be mobilised and quite useful for secular actors.

Each of the cases presented above is determined by larger political issues and power relations. More precisely, in each case, religion is mobilised by secular actors (who become nonreligious in the process) in order to ease the tension and friction that occur in the political field. In the case of the Berlin Wall Memorial, the initial unwillingness of many actors to mark and memorialise the division of the city and the existence of two parts of Germany was countered by the actions of religious actors, who subsequently accepted a role for religion as an invisible manager and organiser that consistently stays in the background and makes space for secular memorialisation practices. In this case, religious materiality is decent and covered, merging with its surroundings. There is no church steeple, no conspicuous cross, and the meeting place of the religious community becomes the garden behind the chapel building, where diversity, including religious diversity, is accepted and accommodated.

In the other two cases, the initiators of the actions were actors motivated by objectives directly related to politics and ideology. In the case of the village churches, one aim was to at least symbolically counteract the long-term social and economic effects of German reunifi-

cation, which resulted in social and economic crises in certain regions. The church buildings become symbols of resistance to the processes of depopulation and impoverishment of the remaining local inhabitants.

In the case of the Garrison Church, many political objectives can be identified, and they vary among the different actors. Even if we assume that right-wing and exclusivist sentiments do not dominate the motivations of those supporting this project, a surely salient aspect of the situation is gentrification and elitist politics. Aesthetics is a political issue, and the rebuilding of the Garrison Church is part of a broader process of increasing the attractiveness of the city for wealthy inhabitants and investors. With regard to the Garrison Church, and in contrast to the situation with the Berlin Wall Memorial, the materiality of religion is not hidden but exposed. The first thing to be rebuilt is the church tower, which codetermines the skyline of the city. The tower of a village church is also often rebuilt first, because a church is not only a meeting place, but also a sign of the presence of the local community.

It is not my aim to provide a typology of the ways in which religious materiality can be present in one of the most secularised regions in Europe. Rather, my aim is to point out several issues that until now have attracted less attention in the study of religion and its relation to secularism as an ideology and secularisation as a process, especially in social anthropology. I also admit that my analysis focuses on religion, which used to be dominant in the region. Although the Evangelical Church is no longer considered a *Volkskirche*, it is still considered "native". I am sure that the dynamics between secular governance and the religious communities that are considered newcomers in the land of Brandenburg look different than those presented in my article. Still, I also think that there is a relative scarcity of ethnographic work focusing on the relations between (previously) dominant religions and secularism in Europe, and this work is intended to contribute to this less studied field.

Firstlydiscussions on the presence of religion in the public sphere usually concern the rights of religious communities and subjects to be treated as equal participants. In my article, we deal with a different but no less potent situation, which is that religion can be mobilised by nonreligious actors without much concern for the rights and wishes of religious communities. Moreover, religious communities can be invited to be present in the public sphere almost against their will and interests. They are seen as repositories of valuable resources that can be mobilised regardless of whether such mobilisation suits them, leaves them indifferent, or provokes their opposition.

Secondly, one of the main threads of the critique of secularisation theory was based on the thesis of the "deprivatisation of religion" (Casanova 1994). In this case, privatisation, considered to be one of the three main forms of secularisation according to Jose Casanova, is reversed: religion becomes a public matter again, and religious actors bring their religious conviction to the public sphere. In the cases described here, however, we deal with neither the privatisation nor deprivatisation of religion. Instead, in the context of confident political secularism, religious symbols, social practices, and materiality are brought out and used in the public sphere, despite the fact that religious convictions concerned with the idea of transcendence and relations with it do not enter this sphere; moreover, it is not even clear whether they are present in the private sphere to any significant extent. Religion is a factor in public debates because of its organisational and material aspects and despite the fact that religious beliefs have considerably diminished or ceased to be a factor for many people.

Within the context of confident political secularism in this particular part of Europe, religion is present in the public sphere because of at least two factors, organisational and material. We have religious organisations whose presence is historically, economically, and administratively well established and socially recognised. They have structures and resources, which afford them a certain stability. Still, their role for believers is diminishing, while their usefulness for secular actors, state and nonstate alike, is recognised. Religion, in this case, is a powerful set of tools that can be mobilised for various kinds of agendas. Religion is present most powerfully in the public sphere. Religious symbols, materiality, and, most importantly, organisations are mobilised to be part of discussions concerning

economic transformations, memorialisation politics, and social issues and relations. However, the decisive voice concerning the role of religion and how those tools are used belongs to secular actors. In the case of church buildings in Brandenburg's countryside, religious materiality and ritualization are mobilised in order to prevent privatisation and promote social binding. In the case of the Garrison Church, religion is used as the justification for a controversial and elitist project. At the Bernauerstraβe, the religious community organises and runs day-to-day activities at the memorial site, at the same time remaining fairly invisible and in the background.

Religion, in this case, is not seen as a problem. For example, Verkaaik and Arab (2016) point out the unidirectionality of power relations between the secular and the religious, as shown in the works of such authors as Talal Asad (2003) and Wendy Brown (2006). Religion is, at best, tolerated, and in this sense, they say, secularism can be called inherently oppressive. Still, Veraaik and Arab also show ethnographically that in some cases, secular governing powers can discipline citizens to accept the law, which actually allows religious communities to discuss issues in public, introduce religious materiality into the public space, and be overtly religious in general.

The cases that I present here follow a similar logical argument concerning political secularism in Europe, but they point in another direction. While Veraaik and Arab focused on the rights of religious minorities and a version of secularism that makes the execution of those rights possible, I point out that religion can become a part of a secular toolkit. Confident political secularism mobilises religion, and while this can be seen as a case of instrumentalization, it also shows that the discursive separation of the secular and the religious can work both ways, making religion—which in the case of Brandenburg and Berlin is generally in a minority position vis-a-vis the secularised population—visible and potent.

**Funding:** This research was funded by National Science Centre (NCN), Poland, grant number Opus 2019/33/B/HS3/02136. The APC was waived.

**Institutional Review Board Statement:** The Study was approved by the NCN Expert Panel.

**Informed Consent Statement:** All participants were duly informed about research protocol and purpose.

**Data Availability Statement:** Not applicable.

**Conflicts of Interest:** The author declares no conflict of interest.

## Notes

[1] The research for this article was funded by grant 2019/33/B/HS3/02136 from the National Science Centre in Poland under the project "Church Buildings in a Secularized Space".

[2] For clarity, in this article I capitalise Church as an organisation and use lowercase church to refer to an edifice. The exception is Garrison Church, which is the name of the place.

[3] https://www.potsdam.de/sites/default/files/documents/statistischer_jahresbericht_2019_landeshauptstadt_potsdam.pdf_online_0.pdf, accessed 2 March 2021.

[4] https://www.ekd.de/ekd_de/ds_doc/Ber_Kirchenmitglieder_2020.pdf, accessed on 17 February 2023.

[5] Stephen Brown, '"Säkulare Realität" Ostdeutschlands ist neue Herausforderung für reformatorische Tradition', *Ökumenischer Rat der Kirchen*, 30 May 2016. Available at: https://www.oikoumene.org/de/news/east-germanys-secular-reality-a-new-challenge-to-reformation-tradition, accessed on 17 February 2023.

[6] Menschen haben vergessen, dass sie Gott vergessen haben.

[7] Martin Kugler 'Menschen haben vergessen, dass sie Gott vergessen haben' *Die Presse* 18.11.2008. Available at https://www.diepresse.com/431291/bdquomenschen-haben-vergessen-dass-sie-gott-vergessen-habenldquo, accessed on 17 February 2023; Karin Vorländer 'Schon vergessen, dass sie Gott vergessen haben¿ Public Forum 23.06.2000. Available at: https://www.publik-forum.de/Publik-Forum-12-2000/schon-vergessen-dass-sie-gott-vergessen-haben, accessed on 17 February 2023.

[8] https://www.ekd.de/staatsleistungen-53875.htm, accessed on 17 February 2023.

[9] https://www.ekd.de/faltblaetter-kirchen-und-kapellen-44500.htm, accessed on 17 February 2023.

[10] See e.g., https://transara.de/, accessed on 17 February 2022.

11    Bern Janowski, chair of the Association, personal communication 05.06.2020.
12    See https://www.altekirchen.de/, accessed on 17 February 2023.
13    https://kirchenbau.ekbo.de/fileadmin/ekbo/mandant/kirchenbau.ekbo.de/netblast/Dokumente_KBA/Kirchen_-_H_auser_
      Gottes_f_ur_die_Menschen__download-datei.pdf, accessed on 17 February 2023.
14    In Prussia, King was used as the official title until 1772.
15    https://www.potsdam.de/potsdamer-mitte, accessed on 17 February 2023.
16    https://www.duden.de/rechtschreibung/Vergangenheitsbewaeltigung, accessed 3 September 2021.
17    https://garnisonkirche-potsdam.de/ueber-uns-1/foerdergesellschaft/, accessed 8 January 2023.
18    https://www.stiftung-berliner-mauer.de/de/gedenkstaette-berliner-mauer/historischer-ort/die-gedenkstaette, accessed on 17
      February 2023.
19    https://gemeinde-versoehnung.de/, accessed on 13 January 2023.

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
