# Peer review of "Church Building as a Secular Endeavour: Three Cases from Eastern Germany"

_religions, doi:10.3390/rel14030287_

Round 1
Reviewer 1 Report
This paper has the goal of investigating the strategies of religious and secular actors towards each other in East Germany. The author presents three case studies: an association restoring church buildings, the case of the rebuilding of the Garrison church, and the Reconciliation Chapel.
I was not convinced by this paper. The three case studies are in themselves quite interesting to read. The three cases are obviously connected and could lead to interesting comparisons and analyses.
But the author just does not seem to know what to do with them. The article does not do what it claims to do. We do not get a typology of possible strategies of religious and secular actors towards each other. Since the author does not treat his/her central question, we do not get any identifiable answer or result. The bottom line of the article seems to be that "religion and religious organisations are not necessarily seen as oppositional to the secular state; there can be a myriad forms of coexistence of secular and religious authorities" (in the abstract). This is hardly an interesting result, and certainly not one that would have needed the analyses given by the author.
My guess is that these three case studies should not be used for this central question. Rather, these three cases suggest a central question such as:
How do church buildings become important for different types of actors in Eastern Germany? Three case studies.
This could lead to an interesting typology and some generalisable insights.
The manuscript should be corrected by a native speaker.
Smaller points:
Abstract:
" one 12 the other investigating strategies that religious actors adopt in a context of political secularism."
Author Response
Response to reviewer 1:
Thank you for your review.
I have had the article corrected by the native speaker after revisions.
Your review was quite different in tone and recommendation from the other two I received. As you seem to be unconvinced by my argument, choice of methodology and also my choice of research question, I decided to follow the advice of the two other reviewers, whose recommendations were closer to my own initial agenda.
I tried to remove all statements from the article which you read as promises that I do not fulfil - I will also change the abstract before publication.
Thank you very much for your insights and I am sorry to disappoint your expectations.
Reviewer 2 Report
I enjoyed reading this very rich ethnographic paper on the various ways the author describes how the religious and the nonreligious are negotiated and used for strategic purposes. It is nicely written and it reads easily. The strength of this article lies in its empirical material: it is an empirical study (three actually) of religious/secular formations and there are not many of such empirical works. I see a lot of this potential in this piece, but first, some serious work is required.
First of all, I think the article requires much more theoretical engagement. There is a whole field of secular studies and not many works from there are discussed. To give just one example: the main argument of this article is quite similar to Asad's point about the secular encompassing/producing the religious/ religious difference. I would love the see a discussion on that interaction, and how it relates to Asad's work. There are many more relevant authors here. To name just a few: Charles Hirschkind, Saba Mahmood, and Shirin Moazami.
Second, I do not think this piece is about political secularism but rather about secularity. If the author wants to stick to the notion of secularism (which sounds strange to me as it is not just about ideology, but about the interaction between ideology and materiality...) , it appears to me to be about the relationship between political and cultural secularism, with the latter being the most prominent in the piece. I think the 2016 article of Oscar Verkaaik and Tamimi Arab in the journal of Muslims in Europe will be a helpful dialogue partner here.
Thirdly, there is no sufficient engagement with the notion of secular materiality. Materiality is mentioned, but not properly discussed, nor is the idea of secular materiality elaborated upon. There have been extensive discussions on this issue, for example about secular buildings and secular bodies. And of course, there is the journal of Material Religion.
Fourth, the empirical parts of the paper are interesting but too detailed I think. They can get more to the point, so there is more room for analysis of the three case studies together. I think the overall argument is sound and well-supported but can be more precise and more carefully developed. OK, we see that nonreligious actors use religious material forms for different purposes, but is there more to be said about those different examples considered together? It appears to relate closely to power, so more needs to be said about that. And what does it imply that this is happening; what are the consequences for society, but also for the study of the secular?
Fifth, the discussion part of this article needs to be extended significantly. Both in the sense of an analysis of the three case studies together (see the previous comment) and in the sense of an engagement with secular studies theory. Currently, it is limited to a very short conclusion and this is by no means enough I'd say. I miss the contribution to secular studies, which is a pity as I think the empirical findings are super interesting.
Minor points:
The third case study reads as a slightly different piece. It should be integrated more appropriately.
The word 'dangerous' on page 7 feels too strong. If the author truly wants to use that word, it should be explained more carefully.
Author Response
Thank you very much for your helpful and insightful review. I tried to address all issues you raised. I also thank you for signing the review report, although I still cannot see your name (journal's policy).
I thank you especially for a reference to Oscar Verkaaik and Tamimi Arab's work - it is extremely helpful.
This article is a response to a specific personal invitation to write with reference to the concept of political secularism. I agree that it is not the obvious choice for my material. But after some reflection I decided that it may provide an interesting avenue for analysis. I introduced now a notion of "confident political secularism". I hope it works well. The article is explicitly about both - secularism and secularity.
I introduced references to some "obvious suspects" in the study of the secular , following your advice, as well as to materiality debate.
After consideration I decided not to shorten the empirical part - the article became much longer but still within the limits set by the journal. I even expanded the third case study.
I extended the conclusion both in terms of theory as well as commentary on the cases.
I want to thank you again for your comments - they were very encouraging, as the field of secular studies is relatively new to me. I will approach it now with more theoretical awarness and boldness.
Reviewer 3 Report
This article presents ethnographic data collected at three sites in Eastern Germany focused upon the restoration of churches in these areas. It demonstrates that religious and secular actors can collaborate for different and/or the same purposes in relation to such buildings in a regional context shaped by the politically secular state; secularization; Landkirche status for these churches, and the legacy of the DDR. It shows that these collaborations vary from locale to locale and that the religious and the secular are enmeshed. I particularly appreciated the attention to actors socially perceived as religious or secular, and the point that political secularism can mean collaboration with religion. This is a nuanced, interesting article worthy of publication.
The article is short, and I’m not sure if this is a requirement of the special issue or not. However, if there is room for more words, I would welcome more detail on the methodology: how were data collected (interviews as well as participant observation)? How were data stored and analyzed? Did the project receive institutional ethical approval? These additions would help the reader understand the research process underpinning the argument more.
Also, a lot of space is devoted to outlining the situation in each locale in relation to the church restoration project, meaning there is less space for expanding upon how religious and secular actors interact in relation to each project. Could more detail be added to support the analysis? The political field is referenced, as are conservative politics specifically, anti-privatisation goals, and a secular desire to defend Christian European identity against Others briefly. Could the interactions of politics, power, and racism be considered further in the discussions of the cases? Religion is really synonymous with Christianity in the article (despite the reference to Jewish groups also receiving state benefits), which makes a lot of sense given the history of the region presented, and this could be made explicit.
Finally, I welcome the use of Quack’s definition of the nonreligious, but the secular is not really defined, and the two terms are still used interchangeably: please clarify the distinction between them and the meaning of the latter.
The English does require revising throughout. I assume the author has English as an additional language and this is not a big issue: it didn’t impact understanding anywhere.
SPECIFIC POINTS
p. 3 Lines 102 – 105 I do not see from the preceding text why it follows that religious actors are not passive and also react to secular initiatives – please explain and justify more.
p. 5 Line 200 so “status of architectural monument” [sic] is another form of official secular state recognition? Add in a note to clarify this- it’s then another example of state involvement with religious buildings.
p. 5 Lines 213 – 218 this fits with Grace Davie’s concept of vicarious religion – worth noting
p. 11 Lines 497 – 498 fits with Jose Casanova’s old public religions argument- also worth citing
Author Response
Thank you very much for your insightful and helpful review. I tried to address all points you have raised.
The article has be thoroughly edited in terms of English language.
I introduced the section on fieldwork (addressing your question on methodology).
The article is now 3000 words longer.
I explained why I focus on Christianity only.
I have not changed the description of the cases much, but I expanded theoretical parts and analysis of the three cases together.
I defined nonreligious with more care.
Thank you very much again, I deeply appreciate your work.
Round 2
Reviewer 1 Report
-